# Low Preoperative Mean Platelet Volume/Platelet Count Ratio Indicates Worse Prognosis in Non-Metastatic Renal Cell Carcinoma

**DOI:** 10.3390/jcm10163676

**Published:** 2021-08-19

**Authors:** Yu-Chiao Lin, Hau-Chern Jan, Horng-Yih Ou, Chien-Hui Ou, Che-Yuan Hu

**Affiliations:** 1Department of Urology, National Cheng Kung University Hospital, College of Medicine, National Cheng Kung University, Tainan 704, Taiwan; whatslin@gmail.com (Y.-C.L.); jan.hauchern@gmail.com (H.-C.J.); 2Department of Surgery, Division of Urology, National Cheng Kung University Hospital Dou-Liou Branch, Yunlin 640, Taiwan; 3Institute of Clinical Medicine, College of Medicine, National Cheng Kung University, Tainan 704, Taiwan; wahoryi@mail.ncku.edu.tw; 4Department of Internal Medicine, Division of Endocrinology and Metabolism, National Cheng Kung University Hospital, College of Medicine, National Cheng Kung University, Tainan 704, Taiwan

**Keywords:** renal cell carcinoma, mean platelet volume, platelet count, ratio, prognosis, oncological outcome

## Abstract

Objectives: Multiple blood parameters are used to determine the prognosis of renal cell carcinoma (RCC). Mean platelet volume/platelet count (MPV/PC) ratio is related to disease progression in various cancers. Our study tried to evaluate the prognostic value of the MPV/PC ratio in RCC patients who underwent surgery. Methods: We retrospectively reviewed 89 patients who underwent radical or partial nephrectomy for RCC in a single institution. Baseline characteristics and MPV/PC ratios were analyzed. The optimal cut-off value of the MPV/PC ratio was determined by a receiver operating characteristic (ROC) curve, and our patients were divided into low and high MPV/PC ratio groups. The Kaplan–Meier survival curve and Cox proportional hazards model were applied for progression-free survival (PFS) and overall survival (OS) analyses. Harell’s C-index was used to compare the prognostic values of the MPV/PC ratio, MPV and PC. Results: Lower MPV/PC ratios were correlated with more advanced tumor stages and worse outcomes. The optimal cut-off value of the preoperative MPV/PC ratio was 0.034 (sensitivity 84.6%, specificity 56.6%). The Kaplan–Meier survival curve revealed that low MPV/PC ratios were associated with worse PFS (*p* = 0.007) and OS (*p* = 0.017). Multivariate analysis showed that low MPV/PC ratios were an independent unfavorable factor for PFS (*p* = 0.044) and OS (*p* = 0.015). Harell’s C-indexes showed that the prognostic value of the MPV/PC ratio was significantly better than MPV and PC (*p* < 0.001). Conclusion: Low MPV/PC ratios are an independent, unfavorable risk factor for disease progression and overall survival in patients undergoing surgery for RCC.

## 1. Introduction

Renal cell carcinoma (RCC) is the most common malignant renal tumor in the world. The prognosis of advanced RCC is poor due to chemotherapy and radiotherapy resistance. Because of the prevalence of RCC and the difficulties in treatment for advanced disease, it’s important to search for clinically available biomarkers to determine its prognosis in daily practice.

In recent years, several parameters based on blood tests, including hemoglobin, neutrophil level and calcium concentration, have shown their value for providing additional prognostic information for advanced/metastatic RCC [1]. Among these, platelet characteristics have been identified as new prognostic factors. Activated platelets may account for oncogenesis and progression of RCC [2]. Increased platelet concentration in blood is also related to unfavorable outcomes in many cancers, mostly reported in ovarian, endometrial and colorectal cancers [3,4,5,6]. However, total platelet count (PC) is dynamic in the human body, balanced by the megakaryopoiesis of bone marrow cells and the consumption rates under different conditions, such as inflammation or microthrombosis formation in tumor microenvironments [7,8,9]. A progressive disease status may be masked by normal PC, due to its effective compensation [10].

Mean platelet volume (MPV), an index of platelet size, can be easily measured by modern blood cell analyzers [11]. It is recognized as a surrogate marker for activated platelets, and is associated with inflammatory conditions or advanced stages of neoplasm [11]. Prokopowicz G et al. reported that among many analyzed parameters to be considered as potential prognostic factors, MPV and PC were the only blood parameters able to predict prognosis of non-metastatic RCC [12]. However, there are no consistent results that demonstrate how MPV indicates the prognosis of cancer [11,13,14,15,16,17,18]. Furthermore, studies have shown a non-linear inverse relationship between MPV and PC, indicating that these two variables should be utilized as a ratio rather than being interpreted alone [19]. The diagnostic value of the MPV/PC ratio has been addressed in several studies [20,21], and our aim is to evaluate whether the MPV/PC ratio could serve as a prognostic factor in RCC patients.

## 2. Materials and Methods

From January 2016 to December 2019, 454 patients received partial nephrectomy or radical nephrectomy in our hospital, and were checked for MPV and PC before the operation. Medical records of these patients were also retrospectively reviewed. We obtained cases with a pathological diagnosis of renal cell carcinoma only. The pathological reports of other cases were upper tract urothelial carcinoma, angiomyolipoma or infection. Then, we excluded children, patients who had platelet disease, history of receiving neoadjuvant medical therapy, acute or chronic inflammatory disease or under anti-platelets medication. Patients who were detected with visceral organ metastases by preoperative imaging technologies, such as CT (computed tomography), MRI (magnetic resonance imaging) or PET (positron emission tomography), were omitted as well. Eventually 89 patients were enrolled in our study (Figure 1). 

Data for age, gender, body mass index (BMI), medical comorbidities, smoking history, Eastern Cooperative Oncology Group *Performance Status* (ECOG PS), laboratory variables, T stage, lymph node (LN) involvement, overall pathological stage and pathological features were collected from their medical charts. Preoperative blood tests were conducted less than one month before surgery. Thrombocytosis was defined as platelet count > 400 (10^3^/uL). Anemia was defined as hemoglobin < 12 (g/dL). Leukocytosis was defined as white blood cell count > 100,000/uL and neutrophilia was defined as neutrophil count > 77,000/uL. Neutrophil-lymphocyte ratio (NLR) was defined as neutrophil count divided by lymphocyte count. Pathological stages of our patients were defined according to the 8th edition of the American Joint Committee on Cancer (AJCC) TNM classification. Postoperative treatment and surveillance were based on the instructions from the National Comprehensive Cancer Network (NCCN) guidelines. Progression-free survival was defined as the time from operation to the day that local recurrence or metastasis was found by image, or to the date of cancer-specific death. Overall survival was defined as the time from operation to the date of death.

Statistical Package for Social Sciences, version 22, for Windows (SPSS Inc., Chicago, IL, USA) was used for analysis. Pearson’s chi-square test and Fisher’s exact test were applied to verify the relationships between baseline characteristics and the MPV/PC ratio. Statistical significance was defined as *p* value < 0.05. According to PFS, the optimal cut-off value of the MPV/PC ratio was determined by the Youden index of ROC curve analysis. The formula of the index was sensitivity + specificity-1. The maximum value of the index was selected as the optimum cut-off point. Kaplan–Meier curves for OS and PFS were analyzed, and comparisons between low and high MPV/PC ratio groups were conducted by log-rank tests. Univariate and multivariate analyses were conducted to evaluate the predictive value of the MPV/PC ratio for PFS and OS. In addition, sensitivity analysis was conducted. Patients with high ECOG PS were excluded, and we conducted univariate and multivariate survival analysis again. The same method was also applied for patients with CKD stage 5. In order to realize the superiority of the predictive ability of MPV/PC ratio, Harell’s concordance index (C-index) of each factor (MPV/PC ratio, MPV, or PC) in addition to a known prognostic factor, pathological stage, was calculated. The C-index value closer to 1.0 indicated that it had better prognostic accuracy.

Furthermore, different studies investigating MPV for cancer prognosis were collected. We listed the cut-off values of MPV in each study, and found the coefficient of variation (CV). We compared it with CV of MPV/PC ratio to analyze the utility of this integrated index.

## 3. Results

In Table 1, among 89 patients, 29 (32.6%) and 60 (67.4%) were female and male, respectively. The median age at time of surgery was 59 years (interquartile range 51.5–66.5). The average of patients’ baseline BMI was 24.9 kg/m^2^. Forty-eight (53.9%) patients received radical nephrectomy and 41 (46.1%) patients received partial nephrectomy as initial treatment for renal cell carcinoma. No positive surgical margin was noted in patients who received partial nephrectomy. Our surgeon preferred to resect more peritumor tissue for an adequate safety margin. Surgical techniques similar to tumor enucleation were not frequently used in our hospital.

Pathological results indicated that 61 (69%) patients were in stage I, 10 (11%) in stage II, 9 (10%) in stage III and 9 (10%) in stage IV. Regarding histopathological type, 62 (70%) were clear cell type, 10 (11%) were papillary type, 14 (15.7%) were chromophobe type, 2 (2%) were MiT family translocation RCC and 1 (1%) was acquired cystic kidney disease associated RCC. The median postoperative follow-up time was 20.7 months (interquartile range 11.9–31.4) for our patients, and the longest follow-up time was 44.3 months.

The mean MPV/PC ratio had a trend of a lower value when patients presented with higher stages or revealed disease progression, including local recurrence or distant metastasis and cancer-specific death (Figure 2). The ROC curve analysis showed that the MPV/PC ratio cut-off value of 0.034 was able to predict the RCC disease progression with a sensitivity of 84.6% and a specificity of 56.6% (AUC = 0.723, 95% CI: 0.554–0.891, *p* = 0.011). According to the optimal cut-off value for the MPV/PC ratio, patients were divided into two groups: 43 (48.3%) patients with MPV/PC > 0.034 and 46 (52.7%) patients with MPV/PC < 0.034. Between these two groups, no significant difference was identified regarding age, gender, BMI, medical comorbidities, smoking history, pathological stage or other histopathological features (Table 1).

The Kaplan–Meier survival analysis indicated that low MPV/PC ratios were significantly associated with a worse 3-year PFS (94% vs. 70%, *p* = 0.007) and OS (93% vs. 74%, *p* = 0.017) (Figure 3). Univariate analysis showed that low MPV/PC ratios clearly correlated with PFS (HR 6.521, 95% CI 1.424–29.854, *p* = 0.016) (Table 2). Other significant parameters related to PFS were overall pathological stage, Fuhrman grade and tumor necrosis. Further multivariate analysis demonstrated that the MPV/PC ratio and the pathological stage were independent predictive factors of a poor PFS (HR 5.391, 95% CI 1.049–27.700, *p* = 0.044; HR 18.261, 95% CI 3.769–88.475, *p* < 0.001). The correlation between the MPV/PC ratio and OS revealed borderline significance (*p* = 0.055) in the univariate analysis (Table 2). Pathological stage and Fuhrman grade significantly affected OS. Multivariate analysis showed a significant effect of the MPV/PC ratio on OS (HR 25.285, 95% CI 1.880–340.158, *p* = 0.015). Age, pathological stage and Fuhrman grade also significantly influenced OS in multivariate analysis. As for sensitivity analysis, three patients with high ECOG PS were excluded, and MPV/PC was still an unfavorable risk factor for PFS in univariate and multivariate analysis. The same results were found if CKD stage five patients were excluded (Appendix A).

More importantly, the C-index of each variable, adding a known prognostic factor, pathological stage, was calculated. The C-index value that was closer to 1.0 indicated that it had better prognostic accuracy. The C-index of the pathological stage alone to predict progression-free survival was calculated through ROC analyses (AUC 0.832, *p* < 0.001). Then, C-index increased after combining MPV with pathological stage (AUC 0.901, *p* < 0.001), or combining PC with pathological stage (AUC 0.876, *p* < 0.001). Moreover, the ability to predict PFS was the best when combining MPV/PC with pathological stage (AUC 0.912, *p* < 0.001). Therefore, the superiority of predictive ability of the MPV/PC ratio over MPV and PC was confirmed.

Additionally, we summarized different studies in Table 3, which shows diverse cut-off values when using MPV to predict cancer outcome. There is a similar cut-off value if we adopt the MPV/PC ratio as an indicator of prognosis in RCC, NSCLC or nasopharyngeal carcinoma. The coefficient of variation (CV) of MPV is 0.12, which is higher than the CV of the MPV/PC ratio, 0.09. This finding is beyond our expectation, and implies the universal application of the MPV/PC ratio to different types of cancer. More studies are needed to verify our finding.

## 4. Discussion

To our knowledge, this is the first study to provide evidence of a robust association between low MPV/PC ratio and high risk of progression or death in RCC patients. In addition, MPV/PC ratio is shown to be an independent prognostic factor in multivariate analysis.

Platelets play a key role in the modulation of cancer progression and angiogenesis [22]. Circulating platelets will aggregate and adhere to the vessel wall endothelium and tumor cells, subsequently covering tumor cells and help to resist the shear force of blood flow and evade the host immune response [23]. Activating platelets will also release microparticles that contain chemokines or growth factors to interact with tumor cells and enhance their proliferation or epithelial-mesenchymal transition [24,25]. Furthermore, tumor endothelial cells will stimulate local platelet adhesion, and, following activation, secrete their angiogenetic or angiostatic content [26]. Platelet production and size can be regulated by cytokines, such as interleukin-6 (IL-6), macrophage colony stimulating factor (M-CSF) or granulocytes colony stimulating factor (G-CSF) [27]. Over expression of IL-6 has been noted in almost all kinds of cancer [28]. These cytokines affect bone marrow cell maturation, megakaryopoiesis and thrombopoiesis [27], which increases PC. On the other hand, substantial consumption of platelets happens in the case of inflammation or cancer. Overall, fluctuation of the PC may compromise its reliability for predicting disease progression [21].

Since larger platelets are more subject to stimulation, the selective depletion of larger platelets might take place in the tumor microenvironment. As a result, the MPV of circulating platelets would be changed. Alteration of MPV value is therefore regarded as an early index of platelet activation [29]. Many researchers have shown their interest in MPV and its relationship with cancer or other inflammatory conditions [11]. MPV is supposed to be increased in infectious diseases, diabetes or obesity [30]. In patients treated in the intensive care unit, elevated MPV has indicated worse survival [31]. However, in some occasions of neoplasm or tissue proliferation, such as nasal polyps, reduction in MPV was noted [32]. Due to the non-linear inverse relationship between MPV and PC, the MPV/PC ratio was preferentially proposed as a predictor in some studies [20,21,33]. Inagaki et al. showed that the MPV/PC ratio was superior to MPV and PC alone for predicting lung cancer survival with multivariate analysis (MPV/PC ratio, HR = 1.668, *p*= 0.0008; MPV, HR = 1.381, *p* = 0.0121; PC, HR = 1.380, *p* = 0.0114) [20]. ROC curve analysis by Cho et al. revealed a better diagnostic performance with the MPV/PC ratio (AUC = 0.884) than with MPV alone (AUC = 0.722) [33].

In our study, we found low MPV/PC ratios were associated with disease progression. Furthermore, the value of the MPV/PC ratio dropped stepwise from early to late stages, which is supported by Zhang et al. [21]. Moreover, the C-index, which added the pathological stage as a known prognostic factor, showed that the AUC of the MPV/PC ratio in our study was 0.912 (*p* < 0.001). This result was better than MPV or PC alone (AUC = 0.901, *p* < 0.001; AUC = 0.876, *p* < 0.001, respectively). There was no significant difference between low and high MPV/PC ratio groups in baseline tumor characteristics, including pathological stage, tumor histology, grade, intratumor hemorrhage and necrosis; or in baseline patient characteristics, including various known factors affecting platelets, such as BMI, hypertension, diabetes mellitus, CKD stages and smoking. In univariate analysis, none of the aforementioned factors influenced PFS or OS. In multivariate analysis, the MPV/PC ratio remained an independent factor that jeopardized PFS and OS. Our results were in line with Inagaki et al., who found a significant reduction in the MPV/PC ratio in patients with advanced non-small cell lung cancer, and that low MPV/PC ratios were an unfavorable prognostic factor for OS. Furthermore, when comparing the value of MPV or MPV/PC ratio in different malignancies, MPV/PC ratio had less of a difference between different cancers that did MPV. It represented that the MPV/PC ratio may be a promising, widely applicable marker candidate among solid cancers.

Therefore, according to previous literature, the possible mechanism that a reduction in MPV/PC ratio correlated with an unfavorable outcome in RCC may be explained as below. Cancer-related inflammation may involve a systemic and local inflammatory response in the tumor, which was shown to facilitate a favorable microenvironment for tumor invasion and metastasis [34]. These processes induced the release and activation of cytokines IL-6, further leading to the enhanced megakaryopoiesis. Meanwhile, cytokines G-CSF and M-CSF secreted from tumor cells also stimulated megakaryopoiesis [35]. The enhanced megakaryopoiesis facilitated an increase in blood PC. Additionally, the consumption of large platelets would be enhanced in inflammatory states, thus reducing MPV [29]. In turn, the resultant drop in MPV/PC ratio would appear in the ongoing inflammation status of cancer.

Most of the prognostic factors to predict the oncological outcome of RCC after surgery include anatomical factors, such as tumor size, renal capsular invasion, venous invasion; or histological factors, such as tumor grade or RCC subtype [36]. However, the data about these factors rely on specimens obtained from surgery, and lack the ability to provide further information if the patient is not suitable or is unwilling to receive operation. On the other hand, the clinical factors to predict survival, such as anemia and neutrophil/lymphocyte ratio (NLR), are still controversial [37]. In patients with metastatic or advanced RCC, the International Metastatic Renal Cancer Database Consortium (IMDC) and Heng’s model segregate patients into three risk categories, and thrombocytosis has been regarded as one of the independent risk factors of poor OS [1]. However, the use of PC to predict oncological outcome in RCC patients after surgery, as occurred in our study cohort, still needs further validation. We herein discuss the possibility to utilize MPV/PC ratio as an improved index, since it shows better predictability than PC alone in this group of RCC patients. CRP and albumin are not included in our routine examination, so analysis of the data is not applicable in our study. Anemia and NLR are not significant prognostic factors in our statistical analysis. In the study by Zhao et al., higher NLR was associated with decreased OS and cancer-specific survival [37]. However, the authors did not discuss the AUC of ROC curve in their study, and the survival difference between high and low NLR group was not very remarkable. The accuracy and prognostic value of NLR for RCC patients are still unclear.

Finally, it is cautiously noted that different histologic subtypes of RCC have characteristic genetic alternations and biologic behaviors. Therefore, our study divided all enrolled RCC, based on the prevalence and clinical outcome, into clear cell RCC and non-clear cell RCC groups. Moreover, clear cell RCC is the most common subtype and has a worse prognosis than papillary RCC, chromophobe RCC and acquired cystic kidney disease associated RCC (mainly occurring in end-stage renal disease patients). As for MiT family translocation RCC, it is quite a rare tumor with highly variable survival outcome.

Our study has limitations. First, it is a retrospective study from a single research center. Because MPV was not utilized in our hospital until 2016, a small sample size and relatively short follow-up period were inevitable. The heterogeneity of our patients and statistical bias, such as overfitting of the model due to the small number of events cannot be fully eliminated. Second, some hemogram indices identified as prognostic factors in other studies are not included in our clinical practice. Third, our patients are mainly Asian. External validation through applying the results to other ethnic groups is lacking. A larger study with a prospective design and longer follow-up is needed. Despite this, our study clearly demonstrates that a low MPV/PC ratio is an independent prognostic factor for RCC, and further investigations are ongoing.

## 5. Conclusions

Low MPV/PC ratio is a significantly independent predictor of a higher risk of progression and worse overall survival. It is more consistent and reliable than MPV or PC alone. These observations may provide us with additional information for treatment plans and decisions.

## Figures and Tables

**Figure 1 jcm-10-03676-f001:**
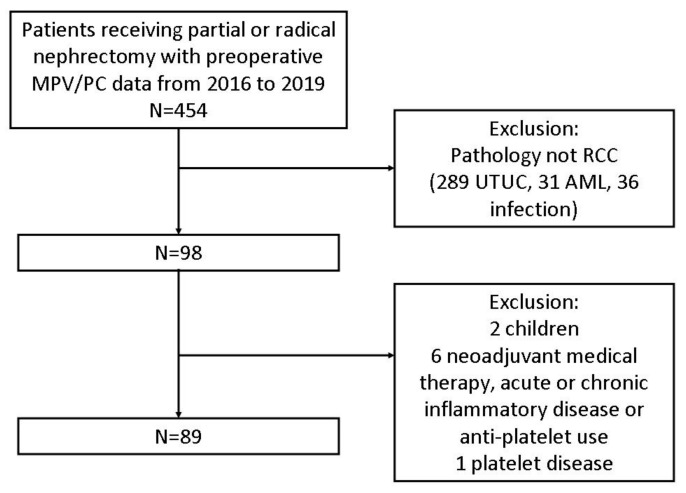
Enrollment of participants of this study. MPV/PC: mean platelet volume/platelet count; RCC: renal cell carcinoma; UTUC: upper tract urothelial carcinoma; AML: angiomyolipoma.

**Figure 2 jcm-10-03676-f002:**
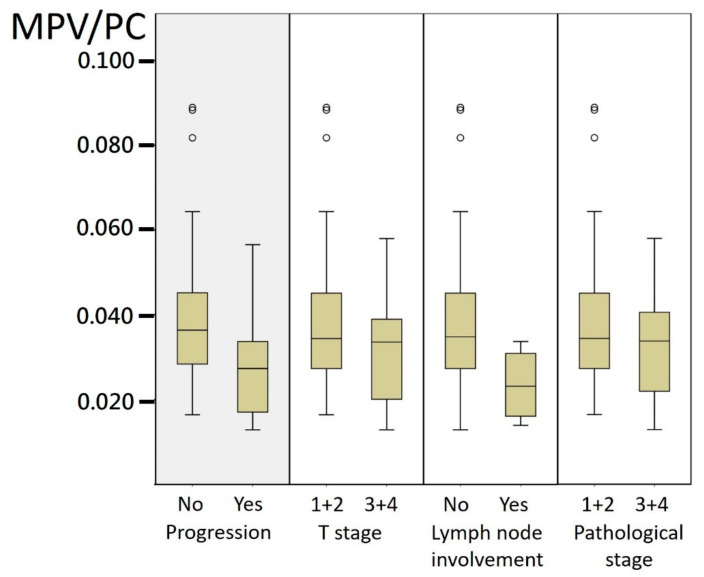
Box plot for mean MPV/PC ratio of different groups. MPV/PC: mean platelet volume/platelet count.

**Figure 3 jcm-10-03676-f003:**
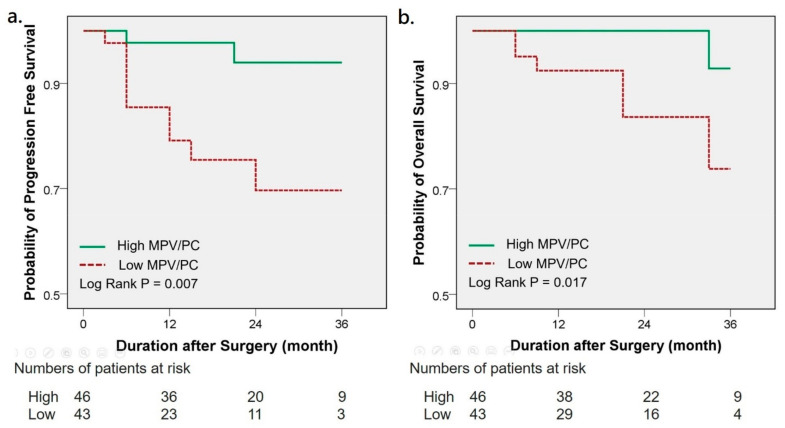
Kaplan–Meier analysis of progression-free survival (**a**) and overall survival (**b**) in RCC patients with high and low MPV/PC ratio. RCC: renal cell carcinoma; MPV/PC: mean platelet volume/platelet count.

**Table 1 jcm-10-03676-t001:** Baseline patient characteristics and tumor characteristics of high and low MPV/PC ratio groups with a cut-off value of 0.034.

	Total (%)	Low (%)	High (%)	*p* Value
Age (years)				0.377
<65	32 (36.0)	30 (69.8)	27 (58.7)	
≧65	57 (64.0)	13 (30.2)	19 (41.3)	
Gender				0.821
Female	29 (32.6)	15 (34.9)	14 (30.4)	
Male	61 (67.4)	28 (65.1)	33 (69.6)	
BMI (kg/m^2^)				0.389
<24	33 (37.1)	18 (41.9)	15 (32.6)	
≧24	56 (62.9)	25 (58.1)	31 (67.4)	
Hypertension				0.821
No	29 (32.6)	15 (34.9)	14 (30.9)	
Yes	60 (67.4)	28 (65.1)	32 (69.6)	
Diabetes mellitus				0.827
No	59 (66.3)	28 (65.1)	31 (67.4)	
Yes	30 (33.7)	15 (34.9)	15 (32.6)	
CKD stage				1.00
1 + 2	71 (79.8)	34 (79.1)	37 (80.4)	
3 + 4 + 5	18 (20.2)	9 (20.9)	9 (19.6)	
Leukocytosis ^1^				0.083
Yes	9 (10.1)	7 (16.3)	2 (4.3)	
No	80 (89.9)	36 (83.7)	44 (95.7)	
Neutrophilia ^2^				0.217
Yes	12 (15.8)	8 (21.6)	4 (10.3)	
No	64 (84.2)	29 (78.4)	35 (89.7)	
NLR				0.360
≧3	33 (42.3)	18 (48.6)	15 (36.6)	
<3	45 (57.7)	19 (51.4)	26 (63.4)	
Anemia ^3^				0.361
No	61 (68.5)	27 (62.8)	34 (73.9)	
Yes	28 (31.5)	16 (37.2)	13 (26.1)	
ECOG PS				0.109
0	86 (96.6)	40 (93)	46 (100)	
1	3 (3.4)	3 (7)	0 (0)	
Nephrectomy				0.289
Radical	48 (53.9)	26 (60.5)	22 (47.8)	
Partial	41 (46.1)	17 (39.5)	24 (51.1)	
Smoking				1.00
No	62 (69.7)	30 (69.8)	32 (69.6)	
Yes	27 (30.3)	13 (30.2)	14 (30.4)	
T stage				0.584
T1 + T2	73 (82)	34 (79.1)	39 (84.8)	
T3 + T4	16 (18)	9 (20.9)	7 (15.2)	
LN involvement				0.051
No	85 (95.5)	39 (90.7)	46 (100)	
Yes	4 (4.5)	4 (9.3)	0 (0)	
Pathological stage				0.600
1 + 2	71 (79.8)	33 (76.7)	38 (82.6)	
3 + 4	18 (20.2)	10 (23.3)	8 (17.4)	
Histology				0.818
Clear cell	62 (69.7)	29 (67.4)	33 (71.7)	
Other	27 (30.3)	14 (32.6)	13 (28.3)	
Fuhrman grade ^4^				0.637
1 + 2	56 (68.3)	26 (65.0)	30 (71.4)	
3 + 4	26 (31.7)	14 (35.0)	12 (28.6)	
Intratumor Hemorrhage				0.389
No	33 (37.1)	18 (41.9)	15 (32.6)	
Yes	56 (62.9)	25 (58.1)	31 (67.4)	
Intratumor Necrosis				0.525
No	41 (46.1)	18 (41.9)	23 (50)	
Yes	48 (53.9)	25 (58.1)	23 (50)	

BMI = body mass index, CKD = chronic kidney disease, NLR = Neutrophil-to-lymphocyte ratio, ECOG PS = Eastern Cooperative Oncology Group Performance Status; ^1^—leukocytosis was defined as white blood cell count > 100,000/Ul; ^2^—neutrophilia was defined as neutrophil count > 77,000/uL. Missing data were found in 13 patients; ^3^—anemia was defined as Hb < 12 g/dL; ^4^—Fuhrman grade was not applicable in 7 patients.

**Table 2 jcm-10-03676-t002:** Univariate and multivariate analyses of progression-free survival and overall survival in patients with renal cell carcinoma.

	Progression-Free Survival				Overall Survival			
	Univariate Analysis		Multivariate Analysis		Univariate Analysis		Multivariate Analysis	
	Hazard Ratio	*p* Value	Hazard Ratio	*p* Value	Hazard Ratio	*p* Value	Hazard Ratio	*p* Value
Age (years)	1.352	0.607	1.324	0.661	4.161	0.089	7.277	0.040 *
≧65 vs. <65	(0.429–4.264)		(0.377–4.645)		(0.806–21.479)		(1.092–48.479)	
Gender	1.393	0.619			0.977	0.978		
Male vs. Female	(0.377–5.148)				(0.189–5.055)			
BMI (kg/m^2^)	0.614	0.399			0.246	0.093		
≧24 vs. <24	(0.198–1.906)				(0.048–1.266)			
Hypertension	1.111	0.864			3.599	0.236		
Yes vs. No	(0.333–3.700)				(0.432–29.975)			
Diabetes mellitus	0.997	0.997			2.475	0.236		
Yes vs. No	(0.300–3.320)				(0.553–11.084)			
CKD stage	1.332	0.667			1.597	0.577		
≧3 vs. <3	(0.360–4.928)				(0.309–8.254)			
Smoking	1.044	0.943			0.972	0.973		
Yes vs. No	(0.314–3.472)				(0.188–5.043)			
Leukocytosis ^1^	2.871	0.114			1.367	0.772		
Yes vs. No	(0.776–10.625)				(0.164–11.363)			
Neutrophilia ^2^	2.261	0.238			3.279	0.193		
Yes vs. No	(0.583–8.772)				(0.548–19.634)			
NLR	1.555	0.467			1.122	0.889		
≧3 vs. <3	(0.474–5.098)				(0.224–5.632)			
Anemia ^3^	2.343	0.141			3.136	0.136		
Yes vs. No	(0.753–7.289)				(0.699–14.069)			
MPV/PC ratio	6.521	0.016 *	5.391	0.044 *	7.946	0.055	25.285	0.015 *
Low vs. High	(1.424–29.854)		(1.049–27.700)		(0.953–66.232)		(1.880–340.158)	
Histology	0.781	0.711			1.004	0.996		
Other vs. Clear cell	(0.211–2.888)				(0.194–5.181)			
Pathological stage	29.882	<0.001 *	18.261	<0.001 *	11.496	0.004 *	5.908	0.045 *
≧3 vs. <3	(6.440–138.665)		(3.769–88.475)		(2.220–59.517)		(1.044–33.437)	
Tumor grade ^4^	3.705	0.027 *	3.166	0.081	6.923	0.022 *	6.755	0.050 *
≧3 vs. <3	(1.161–11.821)		(0.868–11.555)		(1.330–36.030)		(1.004–45.469)	
Intratumor Hemorrhage	0.834				0.425			
Yes vs. No	(0.264–2.631)	0.756			(0.095–1.901)	0.263		
Intratumor Necrosis	4.677		2.038		2.063			
Yes vs. No	(1.023–21.375)	0.047 *	(0.385–10.799)	0.403	(0.40–10.636)	0.387		

BMI = body mass index, CKD = chronic kidney disease, NLR = neutrophil-to-lymphocyte ratio, ECOG PS = Eastern Cooperative Oncology Group Performance Status; ^1^—leukocytosis was defined as white blood cell count > 100,000/uL; ^2^—neutrophilia was defined as neutrophil count > 77,000/uL. Missing data were found in 12 patients; ^3^—anemia was defined as Hb < 12 g/dL; ^4^—Fuhrman grade was not applicable in 7 patients. * indicated *p* value < 0.05.

**Table 3 jcm-10-03676-t003:** The cut-off values of MPV and MPV/PC ratio to determine prognosis in current studies. MPV/PC: mean platelet volume/platelet count.

MPV			
Study	Cancer Type	Cut-off Value (fl)	Coefficient of Variation
Seles et al. [6]	Renal Cell Carcinoma	9.5	0.12138
Tuncel et al. [8]	Colorectal cancer	7.89	
Kumagai et al. [9]	Lung cancer	8.5	
Kilincalp et al. [10]	Gastric cancer	8.25	
Zhang et al. [11]	Esophageal cancer	10.6	
Yun et al. [12]	Renal Cell Carcinoma	7.5	
Gu et al. [13]	Breast cancer	8.45	
**MPV/PC Ratio**			
**Study**	**Cancer Type**	**Cut-off Value**	**Coefficient of Variation**
Zhang et al. [16]	Nasopharyngeal cancer	0.040	0.09876
Inagaki et al. [15]	Lung cancer	0.041	
Our study	Renal Cell Carcinoma	0.034	

## Data Availability

The datasets generated during and/or analyzed during the current study are available from the corresponding author on reasonable request.

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
