# Peer review of "Low Preoperative Mean Platelet Volume/Platelet Count Ratio Indicates Worse Prognosis in Non-Metastatic Renal Cell Carcinoma"

_jcm, 2021, doi:10.3390/jcm10163676_

Round 1

Reviewer 1 Report

The authors evaluate the prognostic value of MPV/PC ratio in renal cell carcinoma patients treated with radical or partial nephrectomy.

1) Unfortunately, the number of sample size is too small. Thus, this paper can't show the definitive conclusion.

2) It is unclear how the cut-off of value of MPV/PC ratio was determined. As the author described, the ROC curve analysis showed that the MPV/PC ratio cut-off value of 0.034 was able to predict the RCC disease extension. What is the RCC disease extension?

3) The presence of visceral and lymph node metastasis should be shown, and it should be included in multivariate models.

4) This cohort included renal cell carcinoma patients treated with radical or partial nephrectomy. However, this cohort appears to included not only localized disease but also locally advanced or metastatic disease, suggesting the heterogeneity of disease extension. This study should focus on either localized disease or advanced disease.

5) Is the information of serum C-reactive protein available in the database? If available, I recommend the authors add it to analyses.

Author Response

Review Journal of Clinical Medicine (jcm-1321914)

Dear Editor and Reviewers,

We would like to thank the Editorial Board of the Journal of Clinical Medicine for their consideration of our manuscript “Low Preoperative Mean Platelet Volume/Platelet Count Ratio Indicates Worse Prognosis in Non-Metastatic Renal Cell Carcinoma” (jcm-1321914) and we would also like to thank the reviewers for their thoughtful comments and helpful suggestions.

We believe that we have satisfactorily addressed the comments of the reviewers, and that addressing these points has indeed significantly improved our manuscript. Below we have provided a point by point response to all the reviewer’s comments.

PART I (Reviewer #1):

The authors evaluate the prognostic value of MPV/PC ratio in renal cell carcinoma patients treated with radical or partial nephrectomy.

  1. The reviewer’s comment: Unfortunately, the number of sample size is too small. Thus, this paper can't show the definitive conclusion.

In response to the reviewer’s concern, that is really our study’s limitation. We made efforts to clarify our data and emphasized that MPV/PC ratio may be as a useful candidate marker for identifying RCC patients who were at risk of disease progression after surgery in clinical practice. Actually, it will be better to collect more patients and further address detailed molecular mechanism for validation of these findings in future.

  1. The reviewer’s comment: It is unclear how the cut-off of value of MPV/PC ratio was determined. As the author described, the ROC curve analysis showed that the MPV/PC ratio cut-off value of 0.034 was able to predict the RCC disease extension. What is the RCC disease extension?

To address the reviewer’s concern, we used ROC analysis to determine the optimal value of MPV/PC ratio according to progression-free survival (PFS). In our study, disease progression was defined as local recurrence or metastasis found by image or cancer-specific death.

Hence, we rephrased “According to PFS, the optimal cut-off value of the MPV/PC ratio was determined by the Youden index of ROC curve analysis.” in the “Patients and methods” section (Page 3, line 96-97) and “The mean MPV/PC ratio had a trend of lower value when patients presented higher stages or revealed disease progression, including local recurrence or distant metastasis and cancer-specific death (Figure 2). The ROC curve analysis showed that the MPV/PC ratio cut-off value of 0.034 was able to predict the RCC disease progression with a sensitivity of …” in the “Results” section (Page 5, line 134-138).

  1. The reviewer’s comment: The presence of visceral and lymph node metastasis should be shown, and it should be included in multivariate models.

In response to the reviewer’s concern, since all patients with visceral metastases had been excluded in our cohort study, we only showed the category “lymph node involvement or not” in Table 1. Since N stage was included as one of the factors of the overall pathological (TNM) stage, and we should not put two confounding factors with obvious interaction into multivariate analysis. Besides, there was no patient with LN metastases in the high MPV/PC group. So, in the multivariate analysis, LN involvement was not added to be analyzed.

  1. The reviewer’s comment: This cohort included renal cell carcinoma patients treated with radical or partial nephrectomy. However, this cohort appears to included not only localized disease but also locally advanced or metastatic disease, suggesting the heterogeneity of disease extension. This study should focus on either localized disease or advanced disease.

In response to the reviewer’s concern and suggestion, our cohort study had excluded patients with visceral metastases detected by the imaging study before surgery. When considering pT stage or pathologic stage, there was no significant difference between high and low value of MPV/PC ratio.

Hence, we inserted a sentence “Patients who were detected visceral organ metastases by preoperative imaging technologies such as CT (Computed tomography), MRI (Magnetic Resonance Imaging) or PET (Positron Emission Tomography), were omitted as well.” in the “Material and methods” section (Page 2, line 72-74).

  1. The reviewer’s comment: Is the information of serum C-reactive protein available in the database? If available, I recommend the authors add it to analyses.

In response to the reviewer’s suggestion, we did not routinely check serum CRP for cancer patients before surgery. Serum CRP test will be considered, except when cancer patients experienced fever episodes before surgery. Thus, we did not have CRP data in our cohort study. 

In conclusion, we again thank the reviewers of our manuscript for their helpful comments which we believe have substantially improved our manuscript. We hope that the reviewers will now find our manuscript suitable for publication in JCM, and eagerly await the response to our revisions.

Reviewer 2 Report

Dear Author,

The manuscript “Low Preoperative Mean Platelet Volume/Platelet Count Ratio Indicates Worse Prognosis in Non-Metastatic Renal Cell Carcinoma” is balanced and well written and supporting other lab studies. I have some comments which need to address:

  1. “Increased platelet concentration in blood is also related to unfavorable outcomes in many cancers, mostly reported in ovarian, endometrial, and colorectal cancers” here the author should provide a reference for each cancer.
  2. Reference no.4 is a review article, it will be better if Author provides original articles.

Renal Clear cell carcinoma in these patients is due to chronic inflammation or because they are sporadic in nature and how the cytokines regulate MPV/PC ratio, this question should be addressed in the discussion section

Author Response

Review Journal of Clinical Medicine (jcm-1321914)

Dear Editor and Reviewers,

We would like to thank the Editorial Board of the Journal of Clinical Medicine for their consideration of our manuscript “Low Preoperative Mean Platelet Volume/Platelet Count Ratio Indicates Worse Prognosis in Non-Metastatic Renal Cell Carcinoma” (jcm-1321914) and we would also like to thank the reviewers for their thoughtful comments and helpful suggestions.

We believe that we have satisfactorily addressed the comments of the reviewers, and that addressing these points has indeed significantly improved our manuscript. Below we have provided a point by point response to all the reviewer’s comments.

PART II (Reviewer #2)

The manuscript “Low Preoperative Mean Platelet Volume/Platelet Count Ratio Indicates Worse Prognosis in Non-Metastatic Renal Cell Carcinoma” is balanced and well written and supporting other lab studies. I have some comments which need to address:

  1. The reviewer’s comment: Increased platelet concentration in blood is also related to unfavorable outcomes in many cancers, mostly reported in ovarian, endometrial, and colorectal cancers” here the author should provide a reference for each cancer.

In response to Reviewer’s suggestion, we have updated references (DOI:10.1016/j.ygyno.2014.01.003; DOI:10.1006/gyno.1998.5078; DOI:10.2217/bmm-2016-0214) in the “introduction” section (Page 2, line 47).

  1. The reviewer’s suggestion: Reference no.4 is a review article, it will be better if Author provides original articles.

In response to the reviewer’s suggestions, we removed the original reference and updated three original articles (DOI: 10.1182/blood.V98.9.2720; DOI: 10.1038/nm973; DOI: 10.1056/NEJMoa1110352) (Page 2, lines 50).

  1. The reviewer’s suggestion: Renal Clear cell carcinoma in these patients is due to chronic inflammation or because they are sporadic in nature and how the cytokines regulate MPV/PC ratio, this question should be addressed in the discussion section

In response to the reviewer’s suggestions, we inserted a short paragraph “Therefore, according to previous literature, the possible mechanism that a reduction of MPV/PC ratio correlated with an unfavorable outcome in RCC may be explained as below. Cancer-related inflammation may involve systemic and local inflammatory response to tumor, which facilitated a favorable microenvironment for tumor invasion and metastasis. These processes induced the release and activation of cytokines IL-6, further leading to the enhanced megakaryopoiesis. Meanwhile, cytokines G-CSF and M-CSF secreted from tumor cells also stimulated megakaryopoiesis. The enhanced megakaryopoiesis facilitated an increase in blood PC. Additionally, the consumption of large platelets would be enhanced in inflammatory states, thus reducing MPV. In turn, the resultant drop in MPV/PC ratio would appear in the ongoing inflammation status of cancer.” in the “”discussion” section (Page10, line 72-82). We also updated three references (DOI:10.1038/nature07205; DOI:10.1073/pnas.1015855107; DOI:10.2174/138161211795049804) in the above paragraph.

In conclusion, we again thank the reviewers of our manuscript for their helpful comments which we believe have substantially improved our manuscript. We hope that the reviewers will now find our manuscript suitable for publication in JCM, and eagerly await the response to our revisions.

Reviewer 3 Report

The work done by Yu-Chiao Lin et al found low mean platelet volume/platelet count ratio (MPV/PC) is an unfavorable risk factor for the prognosis in RCC, the authors retrospectively reviewed 89 RCC patients and analyzed the MPV/PC ratio, determined an optimal value of the ratio which could be used as a biomarker to determine RCC prognosis in the clinical practice. This work is overall very interesting and have a high potential to be applied into the clinical management of RCC patients. The manuscript is well-written, and easy to read, I don't have many concerns except a few points listed below:

  1. Please use PC instead of platelet count after being defined at the first time, eg. page 2 line 51; page 9 line 26 and line 28 and so on.
  2. I would suggest the author consider moving the table 3 and its corresponding discussion to the results part, since this part for me is also analysis of data, but this is just a suggestion, to change or not totally depends on the authors.
  3. Since there are many RCC subtypes that are included in this study, such as clear cell, papillary, chromophobe and so on, although they are all RCC, but their gene mutations, histology, and clinical features are completely different. The authors briefly discussed this point in the last paragraph of discussion part, it would be better if one or two sentences more can be added to make this point clearer.

Author Response

Review Journal of Clinical Medicine (jcm-1321914)

Dear Editor and Reviewers,

We would like to thank the Editorial Board of the Journal of Clinical Medicine for their consideration of our manuscript “Low Preoperative Mean Platelet Volume/Platelet Count Ratio Indicates Worse Prognosis in Non-Metastatic Renal Cell Carcinoma” (jcm-1321914) and we would also like to thank the reviewers for their thoughtful comments and helpful suggestions.

We believe that we have satisfactorily addressed the comments of the reviewers, and that addressing these points has indeed significantly improved our manuscript. Below we have provided a point by point response to all the reviewer’s comments.

PART III (Reviewer #3)

The work done by Yu-Chiao Lin et al found low mean platelet volume/platelet count ratio (MPV/PC) is an unfavorable risk factor for the prognosis in RCC, the authors retrospectively reviewed 89 RCC patients and analyzed the MPV/PC ratio, determined an optimal value of the ratio which could be used as a biomarker to determine RCC prognosis in the clinical practice. This work is overall very interesting and have a high potential to be applied into the clinical management of RCC patients. The manuscript is well-written, and easy to read, I don't have many concerns except a few points listed below:

  1. The reviewer’s comment: Please use PC instead of platelet count after being defined at the first time, eg. page 2 line 51; page 9 line 26 and line 28 and so on.

In response to the reviewer’s suggestion, we have revised them..

  1. The reviewer’s comment: I would suggest the author consider moving the table 3 and its corresponding discussion to the results part, since this part for me is also analysis of data, but this is just a suggestion, to change or not totally depends on the authors.

In response to the reviewer’s suggestion, we moved Table 3 and the discussion part into the “result section”. We additionally inserted a short paragraph “Furthermore, comparing the value of MPV or MPV/PC ratio in different malignancies, MPV/PC ratio had a less difference between different cancers than MPV did. It represented that the MPV/PC ratio may be a promising, widely applicable, marker candidate among solid cancers.” in the “discussion” section. (Page 10, line 68-71).

  1. The reviewer’s comment: Since there are many RCC subtypes that are included in this study, such as clear cell, papillary, chromophobe and so on, although they are all RCC, but their gene mutations, histology, and clinical features are completely different. The authors briefly discussed this point in the last paragraph of discussion part, it would be better if one or two sentences more can be added to make this point clearer.

In response to the reviewer’s suggestion, we inserted sentences “Finally, it is cautiously noticeable that different histologic subtypes of RCC have characteristic genetic alternations and biologic behaviors. So, our study divided all enrolled RCC, based on the prevalence and clinical outcome, into clear cell RCC and non-clear cell RCC groups. Basically, clear cell RCC is the most common subtype and has a worse prognosis than do papillary RCC, chromophobe RCC, and acquired cystic kidney disease associated RCC (mainly occurred in end-stage renal disease patients). As for MiT family translocation RCC, it is a quite rare tumor with highly variable survival outcome.” in the “discussions” section (Page 11, line 103-110).

In conclusion, we again thank the reviewers of our manuscript for their helpful comments which we believe have substantially improved our manuscript. We hope that the reviewers will now find our manuscript suitable for publication in JCM, and eagerly await the response to our revisions.

Round 2

Reviewer 1 Report

The sample size of this study is small, but the finding is interesting. The authors answered my questions properly.